# Temporal and spatial dynamics in soil acoustics and their relation to soil animal diversity

Marcus Maeder[1,2,3]*, Xianda Guo[1], Felix Neff[4,5], Doris Schneider Mathis[4], Martin M. Gossner[3,4]*

1 ETH Zurich, USYS TdLab, Zurich, Switzerland, 2 Institute for Computer Music and Sound Technology, Zurich University of the Arts ZHdK, Zurich, Switzerland, 3 ETH Zurich, Institute of Terrestrial Ecosystems, Zurich, Switzerland, 4 Swiss Federal Institute for Forest, Snow and Landscape Research WSL, Forest Entomology, Forest Health and Biotic Interactions, Birmensdorf, Switzerland, 5 Agroscope, Agroecology and Environment, Zurich, Switzerland

* marcus.maeder@zhdk.ch (MM); martin.gossner@wsl.ch (MMG)

**Data Availability Statement:** All data files are available from https://www.ebi.ac.uk/biostudies/studies/S-BSST799.

## Abstract

The observation and assessment of animal biodiversity using acoustic technology has developed considerably in recent years. Current eco-acoustic research focuses on automatic audio recorder arrays and acoustic indices, which may be used to study the spatial and temporal dynamics of local animal communities in high resolution. While such soundscapes have often been studied above ground, their applicability in soils has rarely been tested. For the first time, we applied acoustic and statistical methods to explore the spatial, diurnal, and seasonal dynamics of the soundscape in soils. We studied the dynamics of acoustic complexity in forest soils in the alpine Pfynwald forest in the Swiss canton of Valais and related them to meteorological and microclimatic data. To increase microclimatic variability, we used a long-term irrigation experiment. We also took soil samples close to the sensors on 6 days in different seasons. Daily and seasonal patterns of acoustic complexity were predicted to be associated with abiotic parameters—that is, meteorological and microclimatic conditions—and mediated by the dynamics of the diversity and activity of the soil fauna. Seasonal patterns in acoustic complexity showed the highest acoustic complexity values in spring and summer, decreasing in fall and winter. Diurnal acoustic complexity values were highest in the afternoon and lowest during the night. The measurement of acoustic diversity at the sampling site was significantly associated with soil communities, with relationships between taxa richness or community composition and acoustic complexity being strongest shortly before taking the soil samples. Our results suggest that the temporal and spatial dynamics of the diversity and community composition of soil organisms can be predicted by the acoustic complexity of soil soundscapes. This opens up the possibility of using soil soundscape analysis as a noninvasive and easy-to-use method for soil biodiversity monitoring programs.

**Funding:** MM - 20'000 CHF - Biovision Foundation
- www.biovision.ch - NO. The funders had no role
in study design, data collection and analysis,
decision to publish, or preparation of the
manuscript.

## Introduction

Ecoacoustics is a relatively young research field in which acoustic indicators of ecological relationships and processes are studied. This is usually done by passively observing a soundscape. The sounds that are audible to humans in a landscape during a certain period are technically recorded and analyzed for their ecological significance [1]. The soundscape under study is normally divided into three classificatory groups of sound sources: biophonies—noises of organic origin; geophonies—noises of inorganic origin; and antropophonies—noises originating from human activity. One main aim of ecoacoustic research is to better understand how these sound sources interact and how they are affected by climatic parameters [2].

Recent technical developments in the field of mobile technologies and micro-engineering have made it possible to investigate a landscape acoustically by using arrays of microphones [3] or independent automatic audio recorders [4] over long periods of time and distributed over large areas. Such arrays may be used to investigate dynamics in the animal communities of a particular landscape. By contrast, conventional animal community inventories based on visual identification or collection are time and resource intensive, and inaccessible areas are difficult to investigate [5]. If recorded soundscapes, especially the acoustic complexity of their biophonies, can be related to biodiversity, it would improve biodiversity monitoring substantially, as it allows continuous monitoring with manageable effort. Ecoacoustic methods are already increasingly used where communities and biodiversity in a given area are to be studied and monitored to provide a basis for environmental policy decisions or for testing conservation efficacy [6]. Thus, acoustic monitoring has great potential for monitoring programs, and biophonic monitoring is increasingly discussed in national and international biodiversity monitoring strategies [7].

Statistical analyses of audio recordings with acoustic indices based on spectral amplitude dynamics are increasingly used to assess and monitor biodiversity in specific habitats and biotopes above ground [8]. Here, different signals correspond to different animal sounds; thus, the measured acoustic diversity/complexity can reflect biodiversity in a local community [9]. Based on broad acoustic measurement spectra that are observed above ground, this method has already been applied successfully [10]. Exemplary work in the field of ecoacoustics has focused on diurnal and seasonal rhythms in soundscapes, for instance, in freshwater ecosystems or rural sanctuaries [11, 12]. Other studies have focused on climatic [13], human-technological [14] and land use influences [15] or on spatiotemporal dynamics in the composition and behavior of animal populations [16]. Ecoacoustic methods have already been used in a wide variety of tropical, montane, and temperate ecosystems—above ground and underwater [17]. However, some white spots remain on the global sound map.

One of these white spots is soil ecosystems. Only a few studies have addressed ecoacoustics in soils, and these mainly focused on physical phenomena such as the movement of water fronts through the pore system [18] or changes in the structure of the soil matrix [19]. Other initial studies have focused on soil biophonies and attempted to detect soil pests acoustically [20], have investigated acoustic emissions of insects that use the substrate (plant surfaces and soils) for vibratory communication [21] or addressed the biological activity related to earthworms [22]. By contrast, audio recordings belowground with a restricted analyzable spectrum and focusing on a broad spectrum of soil animals have never been analyzed before. In addition, there have been no attempts to analyze soil soundscapes across temporal and spatial scales and to relate them to soil animal biodiversity, although the potential of such research has already been pointed out [23] and first studies on individual soil animal species underline this (see references 34/35). Comparisons of acoustic diversity with conventional species counts in local soil animal communities do not exist to date.

The composition and behavior of animal communities in soils are very difficult to study directly because the species live in an inaccessible, complex matrix of mineral, organic, and liquid components. So far, the activity and behavior of individual groups of soil organisms have been mainly traced indirectly, for instance, by their decomposition performance [24] or by measuring soil respiration [25]. Direct measurements of the composition of a local community have been restricted to cumbersome, elaborate, and destructive methods, such as pitfall trapping or taking soil samples with soil cores of different sizes and extracting the animals using Winkler, Berlese, Tullgren, or Macfadyen methods [26]. By contrast, passive acoustic observation has clear advantages: the habitats remain undisturbed, animals do not have to be collected and killed, and it allows for continuous monitoring across spatial and temporal scales.

Transferring recording technologies developed for aboveground and underwater setups to soil systems is challenging. For investigations of acoustic emissions of the local fauna in both surface and underwater systems, microphones or hydrophones are typically used [27]. These acoustic sensing devices are complemented by amplifier circuits that make it possible to detect and record sounds inaudible to the human ear [28]. Such recording technologies, however, have limitations in soils. To date, the sounds of functionally important microorganisms (fungi, algae, bacteria, protozoa), for instance, cannot be detected; these organisms are too small and their acoustic emissions too weak to propagate over large distances in the soil matrix. The limiting factors in soil acoustic transmission consist mainly of damping and reflection effects [29].

We used the recording technology that was developed for soils in a preliminary study [30] to detect the acoustic activity of the soil mesofauna and macrofauna. These organisms are primary decomposers and their predators [31], including arthropods and annelids, and live in the litter as well as in the uppermost organic layers [32]. These important functional groups not only produce movement and feeding noises but also use the soil matrix for acoustic/vibratory communication [33]. Thus, it is assumed that a higher diversity of soil animals produces a more complex soundscape that can be detected by acoustic sensors.

Little is known about daily or seasonal behavioral patterns and dynamics in the composition of the soil fauna and its local communities. There have been first attempts to relate subterranean acoustic activity patterns of individual species to abiotic and biotic factors [34, 35]. However, to our knowledge, the present study is the first to investigate the spatial and temporal dynamics of sounds produced by soil mesofauna and macrofauna at the community level and to relate them to abiotic/microclimatic parameters. We addressed the question of whether acoustic complexity measurements can be applied to belowground systems to track diurnal and seasonal dynamics in soundscapes and to relate them to the dynamics of microclimatic conditions, soil biodiversity, and community composition.

## Materials and methods

### Study site

Our study was carried out in the Pfynwald forest in the Canton Valais (46˚ 18' N, 7˚ 36' E, 615 m a.s.l.), Switzerland. With an average temperature of 9.2˚C and a mean annual precipitation of 657 mm (1961–1990, WSL 2018), the forest is located in one of the driest inner alpine valleys of the European Alps, where increasing drought periods lead to increased tree mortality [36].

The study was set up within the irrigation experiment initiated in 2003 by the Swiss Federal Research Institute for Forest, Snow, and Landscape Research WSL [37] (Fig 1), targeted at obtaining a high spatial variability in soil microclimatic conditions. Every year between April and October, half of the plots (irrigation treatment) were irrigated by a sprinkler system providing an additional 700 mm of water annually, and the other half served as control. Parts of the irrigation plots' irrigation were stopped in 2013. To cover different microclimatic

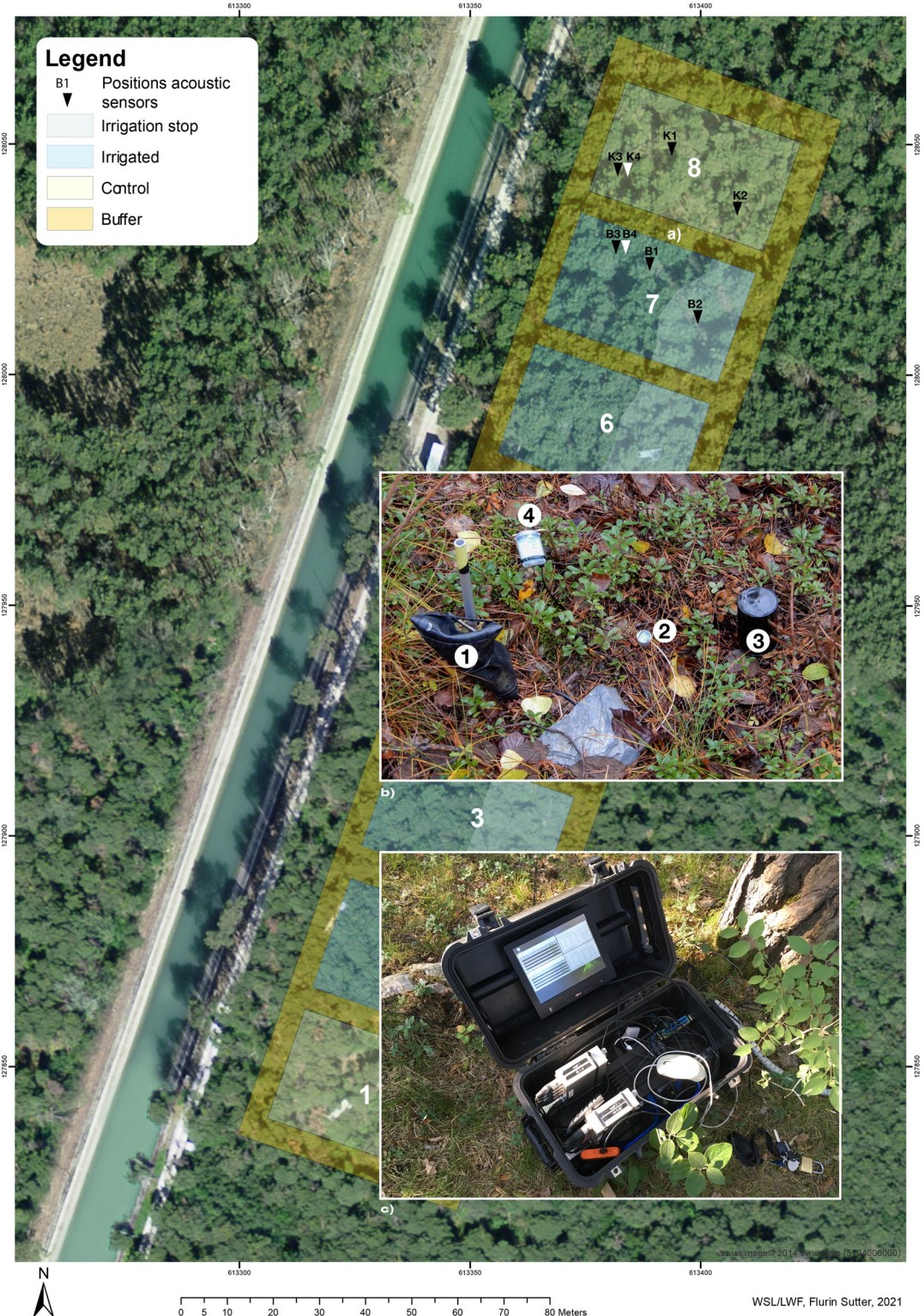

**Fig 1. a)** Acoustic sensors marked with black triangles are located at plots 7 (irrigated/irrigation stop) and 8 (control) of the irrigation experiment in Pfynwald, Valais, Switzerland; white triangles are control sensors in the air. **b)** Local setup of acoustic and microclimatic sensors: 1 acoustic sensor; 2 preamplifiers; 3 soil moisture and temperature; 4 surface temperature and light intensity. **c)** Case with computer and audio interfaces. Adapted from the map of the irrigation project Pfynwald (Orthomosaic © 2019 WSL, swissimage © 2014 swisstopo).

conditions, we distributed our acoustic sensors in plots that were irrigated during the whole period since 2003, plots where irrigation was stopped, and control plots that received no irrigation.

## Assessment of the soundscape

The acoustic measurements began in March 2018. Equipment measuring the acoustic activity of the soil fauna was installed across the three treatments (irrigation/irrigation stop/control, Fig 1A). Three acoustic sensors were distributed randomly in the control (K1–K3), two in the irrigation, and one in the irrigation stop treatment (B1–B3) at distinct distances from the recording computer (Fig 1C), and two additional sensors were mounted to distinguish air-borne and soil sounds (K4, B4).

The recording equipment consisted of an array of specially designed contact microphones (see S1 Fig), covering a soil depth of approx. 10 cm and a volume of approx. 1000 cm³. The signal from the sensors was amplified by +40 dB (a factor of 100) using modified hydrophone preamplifiers by Avisoft, Berlin. Signals from both plots were digitized by two 4-channel audio interfaces (Avisoft *UltraSoundGate 416h)* and recorded on a minicomputer at 10-min intervals (20-s recordings) using Avisoft's *Recorder* software (Fig 1C). The loss of signal strength due to long audio cables (25 m) was compensated by +18 dB in the internal amplifiers of the audio interfaces, and the captured signals were recorded with a high-pass filter of 0.05 KHz (512 taps, filtering low-frequency noise out) and a sampling rate of 50 KHz. The whole system was powered by a fuel cell.

## Assessment of microclimatic conditions

To relate the soundscape, especially its biophonies, to microclimatic conditions, additional local measurements of soil moisture and temperature at a depth of 10 cm were taken during the 2019 growing season (June 15 until July 19) in 10-min intervals with PlantCare *MiniLoggers* (PlantCare AG, Russikon, Switzerland) that were installed next to the acoustic sensors. This time period was chosen because we expected highest activity of soil fauna during this period based on data from 2018. At the same measuring points, we installed Hobo *Pendant*® sensors/loggers (Onset, Bourne, MA/USA) that measured temperature and light intensity on the soil surface (schematic of the setup: S2 Fig). This was done because our acoustic measurements during the first period in summer 2018 indicated a possible relationship between acoustic emissions and sun radiation, that is, the heating of the soil surface. Since the measurement of soil microclimate was not reliable for one sensor, microclimate data was only available for five acoustic sensors.

## Assessment of soil fauna

To relate the recorded biophonies to biodiversity, once per season, a soil sample was taken at all measuring points. Sampling was conducted at six dates between April 2018 and July 2019 (for dates, see S2 Table). For this purpose, a sample cylinder of 5 cm in diameter and 15 cm in length was driven into the soil with a hammer (see S3 Fig) in a circular scheme with a radius of 50 cm with the acoustic sensor in the center (S2 Fig). The samples were cooled and taken to the laboratory, where the soil animals were extracted using the Berlese method with 40 W bulbs for 14 days [38]. Extracted animals were collected in 70% ethanol and stored in vials until further identification (S4 Fig).

Extracted individuals were subsequently sorted into the following taxonomic groups and counted: Acari, Araneae, Chilopoda, Coleoptera, Collembola, Dermaptera, Diplura, Diplopoda, Diptera, Gastropoda, Haplotaxida (Enchytraeidae), Hemiptera (Auchenorrhyncha,

Heteroptera, Sternorrhyncha), Hymenoptera (Formicidae, remaining groups), Isopoda, Lepidoptera, Mecoptera, Nematoda, Neuroptera, Opiliones, Opisthopora (Lumbricidae), Pauropoda, Protura, Psocoptera (Liposcelididae, remaining groups), Symphyta, and Thysanoptera.

## Analyses and statistics

All statistical analyses were conducted in R version 4.0.2 [39].

**Acoustic analyses.** In the first step, the qualitative/spectral texture of the acoustic material was investigated. This was done by listening to randomly selected audio recordings of different times of days, months and seasons. If significant patterns (signals with an amplitude well above the noise floor or with peculiar frequency signatures and dynamics) were heard, spectrograms of the corresponding recordings were created in Adobe Audition [40], and the sound sources were subjected to a closer examination regarding their spectral and temporal structure as well as their spatial distribution (Fig 2). On the one hand, this procedure also allowed us to identify whether the signals occurred in the soil substrate or were airborne (the latter would indicate crosstalk between channels). On the other hand, spectral analysis made it possible to gain insight into the distribution of the signals in the frequency spectrum and to accordingly set filters for excluding unwanted environmental noise before the material was statistically analyzed. In our analysis, we focused mainly on biophonies and geophonies. Anthropophonies also occurred in soils (see S5 Fig) but were irrelevant to our study.

The complexity of a soundscape can be described by different acoustic indices. Hence, in the second step, a pre-test was conducted to decide on the acoustic index used in the study. The following indices were calculated based on audio data recorded in June 2018: acoustic complexity index (ACI), acoustic evenness index (AEI), acoustic richness (AR), and median amplitude envelope (M) [41, 42]. The results of the pre-test showed that no index other than the ACI varied over time, and it was even hard to distinguish the daily irrigation periods in the graphs of AEI and AR. Moreover, ACI performed best at resolving daily and seasonal patterns. For details of the pre-test (see S6 Fig).

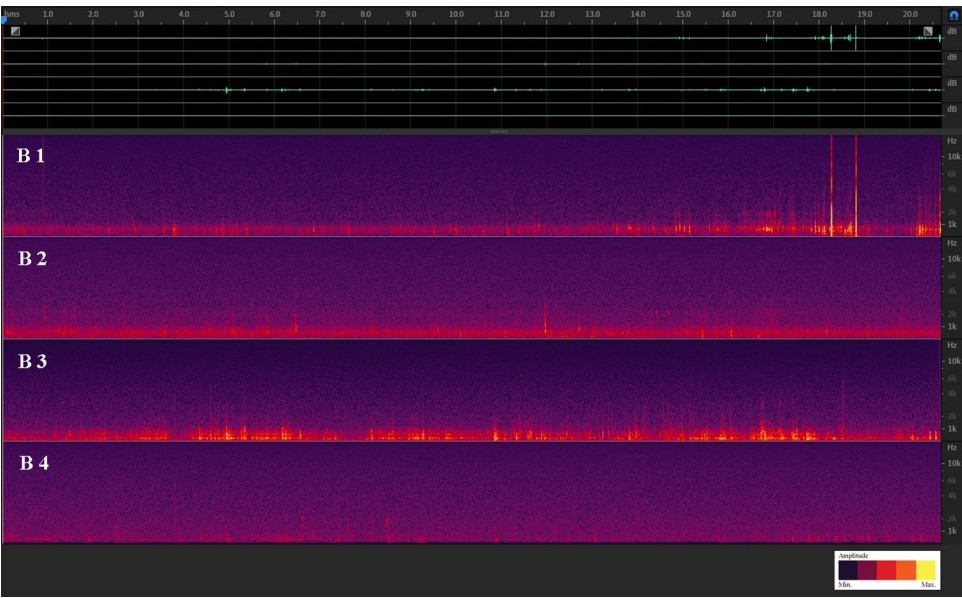

**Fig 2. Spectrogram of a 4-channel recording from the Pfynwald soil (sensors B1–4, irrigated plot).** Bright green waveforms on top: Display of the audio waveforms channels 1–4. Red/purple colored channels: Spectrograms channels 1–4, aligned from top to bottom. x axis: time, y axis: frequency. Channels 1–3 indicate sensor locations B1–B3 in Fig 1A). B4 is a control sensor in the air next to sensor B3. Channel 1 shows the movement sounds of a soil animal.

Calculations using ACI focus on the dynamics of the 'amplitude envelope in single frequency bands and are therefore less sensitive to constantly present sound sources. ACI directly measures the variations in intensities within a recording. First, it calculates the absolute difference between adjacent values of intensity:

$$d_k = |I_k - I_{(k+1)}|.$$

The summation of the $d''$ encompasses the changes in the temporal step of the recording:

$$D = \sum_{k=1}^{n} d_k.$$

To obtain the relative intensity and reduce the influence of the distance between the microphone and the vocalizing organisms, the result $D$ is then divided by the total sum of the intensity values:

$$ACI = \frac{D}{\sum_{k=1}^{n} I_k}.$$

These simple formulas contribute to the wide utility of ACI in dealing with acoustically complex environments. We calculated the ACI from the audio files recorded by the acoustic sensors by adapting Pieretti's method [43] to our data and using the function *acoustic_complexity* in the 'soundecology' package [44].

The ACI was applied to the 10 s/10 min interval recordings. The window length was set at 512, meaning the signal sequence would be divided into a matrix of 256 frequency bands and 2020 time blocks (the actual recordings were marginally longer than 20 s). ACI within the 20-s recordings was calculated with a window length of 1 s. Instead of calculating the sum, we took the average so that the resulting values were within the range of 0 and 1. Only ACI values between 0.55 and 0.90 were included in the analyses. Values lower than 0.55 proved to be unreliable because they lay in the range of constant background noise, which is caused by amplification. Values higher than 0.90 represent direct noise from rain or irrigation events, which were not the focus of the analyses.

**Measures of animal diversity and community composition.** Total abundance and four diversity metrics, as well as community composition, were used as soil community attributes. To calculate diversity, we used the abundances of the different taxonomic groups described above. We calculated Hill numbers qD to quantify diversity in units of equivalent numbers of equally abundant taxa by increasingly weighting abundance with the order of diversity q. We used $^0D$ corresponding to taxa richness, $^1D$ to the exponential of Shannon's entropy (henceforth "Shannon"), and $^2D$ to the inverse of Simpson's concentration (henceforth "Simpson") [45]. Calculations were performed with the 'vegan' package [46]. Community composition was calculated by non-metric multidimensional scaling (NMDS) using the *metaMDS* function based on Bray–Curtis distances in 'vegan'. The first two axes were used for further analyses.

**Relation of ACI to daytime, season, and microclimate.** Time series analyses were used to analyze the effects of season and daytime, as well as of microclimate variables, on temporally resolved ACI values. Models were implemented as Bayesian hierarchical models with a gamma-distributed response variable and an autoregressive model for the autocorrelation structure. Prior to the analyses, 0.55 was subtracted from all ACI values to create a response variable that better met the distributional assumptions. All numeric predictor variables were

scaled to mean 0 and SD 1 prior to analyses. The linear predictor $\eta$ had the form:

$$\eta_{it} = \alpha_i + \boldsymbol{X_{it}}\boldsymbol{\beta} + \sum_{l=1}^{L}\varphi_l \cdot \frac{1}{e^{y_{t-l}}} \tag{1}$$

where $\alpha_i$ is the random intercept for sensor $i$, $\boldsymbol{X_{it}}$ contains the predictor variables at time $t$, $\boldsymbol{\beta}$ contains the model parameters for the predictor variables, $L$ is the order of the autoregressive model (i.e., maximum lag that was considered), $\varphi_l$ is the coefficient of the autoregressive model for lag $l$, and $y_{t-l}$ is the observed response variable at time $t$–$l$. The linear predictor was linked to the response $y$ with

$$y \sim Gamma(k, e^{\eta}) \tag{2}$$

where $k$ is the shape parameter of the gamma distribution. Random intercepts and model coefficients for the sensors were modeled as

$$\alpha \sim N(\mu_{\alpha}, \sigma_{\alpha}^2) \tag{3}$$

$$\beta_j \sim N(0, \sigma_{\beta_j}^2) \tag{4}$$

where $\mu_{\alpha}$ is the global mean of the linear predictor and $\sigma$ denotes the standard deviation for the parameters. Weakly informative priors were used for standard deviations, autoregressive model coefficients and the shape parameter (S3 Table). All hierarchical models were implemented in Stan and run through the R interface 'rstan' (Markov chain Monte Carlo settings: four chains with 2000 iterations each, including 1000 warm-up iterations) [47]. The model output was then used to predict ACI values for different scenarios of predictor variable levels.

Two different data sets were used to answer different study questions. First, to analyze the effect of season and daytime on ACI, data covering 466 days (June 2018 to July 2019) were used. To reduce temporal autocorrelation, data were averaged for intervals of 6 h (12 p.m.–6 a.m., 6 a.m.–12 a.m., 12 a.m.–6 p.m., 6 p.m.–12 p.m.). As predictor variables, daytime (the four intervals indicated before) and season (winter, spring, summer, and fall) and their interaction were included. Second, to analyze the effect of microclimates on ACI, data from the 35 days during the growing season 2019, for which microclimatic data was available, were used. Given that the microclimate was expected to have more immediate effects on ACI, the data were not averaged but modeled with a temporal resolution of 10 min. As predictor variables, soil moisture, soil temperature, and surface temperature were used. As the relation of ACI to soil moisture may be non-linear, the quadratic term of soil moisture was also included in the model.

Further, strong heating might affect ACI through physical processes, that is, geophonies such as evaporation of soil pore water near the surface (see lab experiment, S9 Fig). Thus, positive surface temperature change within 30 min prior to the measurement was included as an additional predictor variable. ACI might also show a lagged response to soil moisture, soil temperature, and surface temperature, prompting us to analyze cross-correlation prior to hierarchical modeling. Highest cross-correlation across sensors for pre-whitened, differentiated time series was found for surface temperature at lag –1 (S10 Fig). There was generally a short lag of a few minutes between measurements of ACI and microclimate variables, which indicated that ACI showed direct, unlagged responses to microclimate change. Higher negative lags would indicate an effect of future microclimates on ACI, which is not plausible. Thus, all microclimate variables were adjusted to the lag –1. To choose the maximum lag $L$ for the autoregressive part of the two models, simple linear mixed-effect models with a log-transformed response variable were used on the data prior to implementing the hierarchical models. From the plot of the autocorrelation function of the model residuals, the maximum lag $L$ was chosen at a reasonable local maximum of the

autocorrelation function (8 for the first model, 10 for the second model; S11 Fig). Linear mixed effect models were implemented with the package 'nlme' [48].

**Seasonal effects on soil community attributes.** To test for the effects of season (spring, summer, fall, winter) on soil communities, we conducted linear mixed effects models (function *lme*, package 'nlme') with season as fixed and sensor as random effect. We modeled the response of each of the four community attributes (abundance, three diversity metrics) separately. Based on inspections of diagnostic plots, abundance and taxa richness were sqrt-transformed prior to analyses. ANOVA tables of the model results are presented.

Differences in communities between samplings were illustrated, showing the first two NMDS axes, and the effects of season on the community composition were tested using PERMANOVA with 1000 permutations using the *adonis* function in 'vegan'. The repeated measures per plot were considered by using a sensor in the *strata* argument.

**Effects of soil community attributes on acoustic measurements.** We tested the effects of abundance and diversity on ACI using linear mixed-effects models (function *lme*, package 'nlme') with one of the community attributes (abundance, three diversity metrics, two NMDS axes) as fixed effects and the plot as random effects. We modeled the response of three different ACI resolutions separately—ACI directly before sampling, 1h mean around sampling, and 24h around sampling. To make the model estimates comparable, abundance and diversity measures were standardized to zero mean and unit variance using the *decostand* function in the 'vegan' package. Model performance was described as conditional and marginal $R^2$ using the *r2* function in the 'performance' package [49]. Predicted relationships between community attributes and ACI are presented as effect plots using the *predictorEffect* function in the package 'effects' [50, 51].

## Results

### Diurnal and seasonal patterns of acoustic complexity

The ACI was observed to be highest in spring and summer, particularly around noon (Fig 3A). The hierarchical time-series model predicted that the ACI in spring and summer was, on average, 0.0143 (0.0028, 0.0285; 95% highest density interval) higher than in winter and 0.0082 (-0.0016, 0.0179) higher than in fall (Fig 3B). There was a strong diurnal pattern with the lowest values during night (6 p.m.–12 p.m.) and in the early morning (12 p.m.–6 a.m.), and the highest values in the afternoon (12 a.m.–6 p.m.) (Fig 3B). This pattern was particularly evident in spring and summer, leading to the largest differences in ACI between seasons in the afternoon, with predicted values in spring and summer being, on average, 0.0261 (0.0224, 0.0301) higher than in winter and 0.0155 (0.0117, 0.0193) higher than in fall.

### Relationship between acoustic complexity and microclimatic conditions

Analyses of the relationship between different microclimatic variables and ACI showed a particularly strong positive effect of a positive surface temperature change on the ACI (heating) (Figs 4 and 5). For example, heating of 5˚C in the last 30 min was predicted to increase the ACI values by, on average, 0.0157 (0.0139, 0.0173). Additionally, the ACI values were observed to be higher with higher surface temperatures but lower with higher soil moisture (Fig 5). There was no significant non-linear signal of soil moisture on the ACI. Furthermore, soil temperature had a weak negative effect on the ACI.

### Links between ACI and soil animal communities

We sampled a total of 7,538 individuals of 21 taxa. The most abundant taxa were Acari (63% of all individuals), followed by Diptera, which emerged as adults (28%), and Collembola (7%).

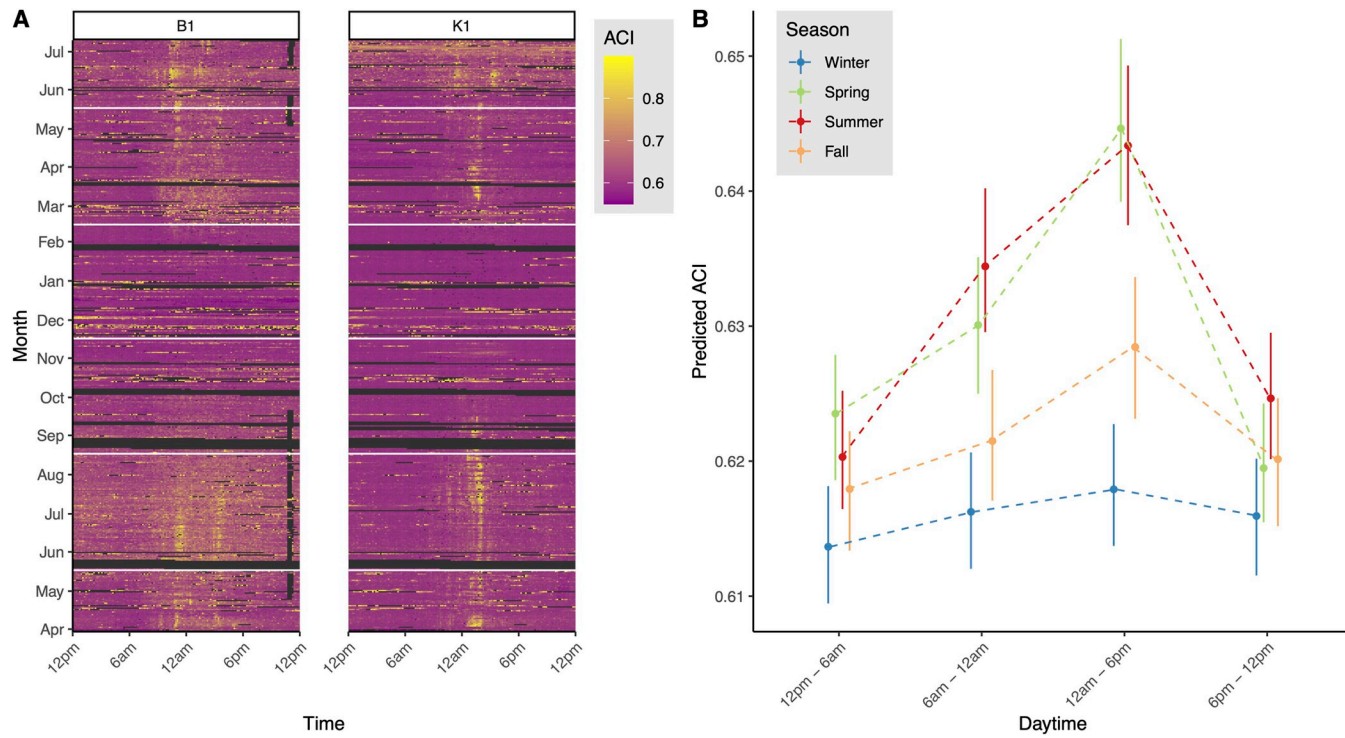

**Fig 3. a)** Observed ACI values from two sensors (B1, K1) over the study period. Each day is represented by a horizontal line, with colors indicating the ACI at a certain time (measured every 10 min). White horizontal lines indicate transitions between seasons. Dark grey areas show missing or excluded data. Data on rain and irrigation were excluded, which is, for example, evident for the irrigation events at sensor B1 in the evenings in spring and summer. **b)** Predicted values from hierarchical models analyzing the effect of daytime, season, and their interaction on ACI. Colors indicate seasons. Points show the highest maximum a posteriori estimates, and error bars show the 95% highest density intervals. Data were aggregated at intervals of 6 h prior to analyses.

For a full list of sampled taxa, see S2 Table. Abundance and diversity of soil animal communities were not affected by season; however, there was a difference in diversity when dominant taxa were more strongly weighted (Simpson; Table 1). This suggests that the dominance of particular taxa but not taxa richness shifted among seasons. Communities based on taxa composition significantly differed among seasons (PERMANOVA p <0.01) (Fig 6).

Our models explained up to 41% of the variation in ACI (taxa richness directly before sampling), and up to 17% (taxa richness directly before sampling) of the variation were uniquely

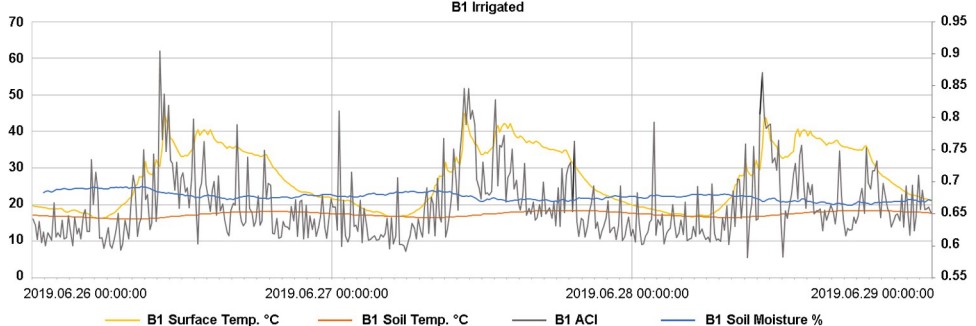

**Fig 4. Example of diurnal dynamics in microclimatic parameters and ACI during three days (26.06.2019–28.06.2019) for sensor B1.** Temperature and soil moisture are represented on the left axis, whereas ACI is represented on the right axis. Soil temperature and moisture were measured at 10 cm depth. These three days show a general pattern that may also be found in other periods (S12 Fig).

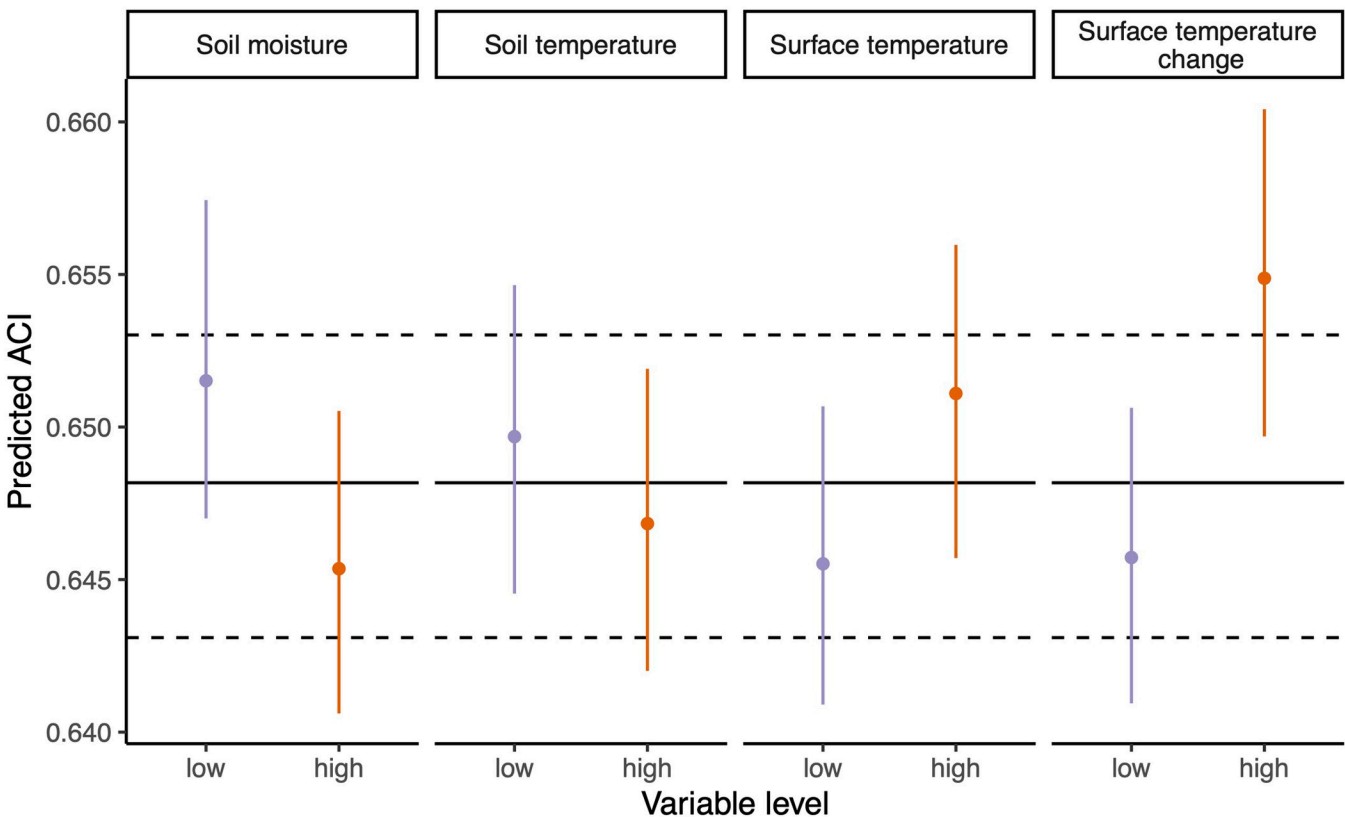

**Fig 5. Predicted values from hierarchical models analyzing the effect of microclimate on ACI.** Four microclimate variables were included in the models. Surface temperature change denotes the positive temperature change 30 min prior to the ACI measurement. For each variable, predictions are shown for a scenario at a low level (mean−1SD) and at a high level (mean + 1SD). For surface temperature change, the low-level scenario was set to 0 degrees temperature change to keep scenarios realistic. For soil moisture, a quadratic term was included in the model to account for a potential non-linear effect. Points show the highest maximum a posteriori estimates, and error bars show the 95% highest density intervals. Lines in black show the global mean, with its 95% highest density interval.

explained by soil community attributes (Fig 7A and 7B). Taxa richness best predicted ACI, followed by taxa composition. The ACI directly before sampling the soil community was best predicted by taxa richness. Both taxa richness and taxa composition were positively associated with ACI directly before sampling (Fig 7C and 7D).

**Table 1. Results of linear mixed-effects models on the effect of season (spring, summer, fall, winter) on the abundance and diversity of soil communities sampled from soil cores.**

|  | numDF | denDF | F-value | p-value | Cond. $R^2$ | Marginal $R^2$ |
|---|---|---|---|---|---|---|
| **Abundance** |  |  |  |  | 0.110 | 0.110 |
| Intercept | 1 | 27 | 123.284 | <0.001 |  |  |
| Season | 3 | 27 | 1.439 | 0.253 |  |  |
| **Taxa richness** |  |  |  |  | 0.072 | 0.017 |
| Intercept | 1 | 27 | 1677.264 | <0.001 |  |  |
| Season | 3 | 27 | 0.218 | 0.883 |  |  |
| **Shannon** |  |  |  |  | 0.239 | 0.164 |
| Intercept | 1 | 27 | 232.815 | <0.001 |  |  |
| Season | 3 | 27 | 2.514 | 0.080 |  |  |
| **Simpson** |  |  |  |  | 0.314 | 0.197 |
| Intercept | 1 | 27 | 314.653 | <0.001 |  |  |
| Season | 3 | 27 | 3.358 | 0.033 |  |  |

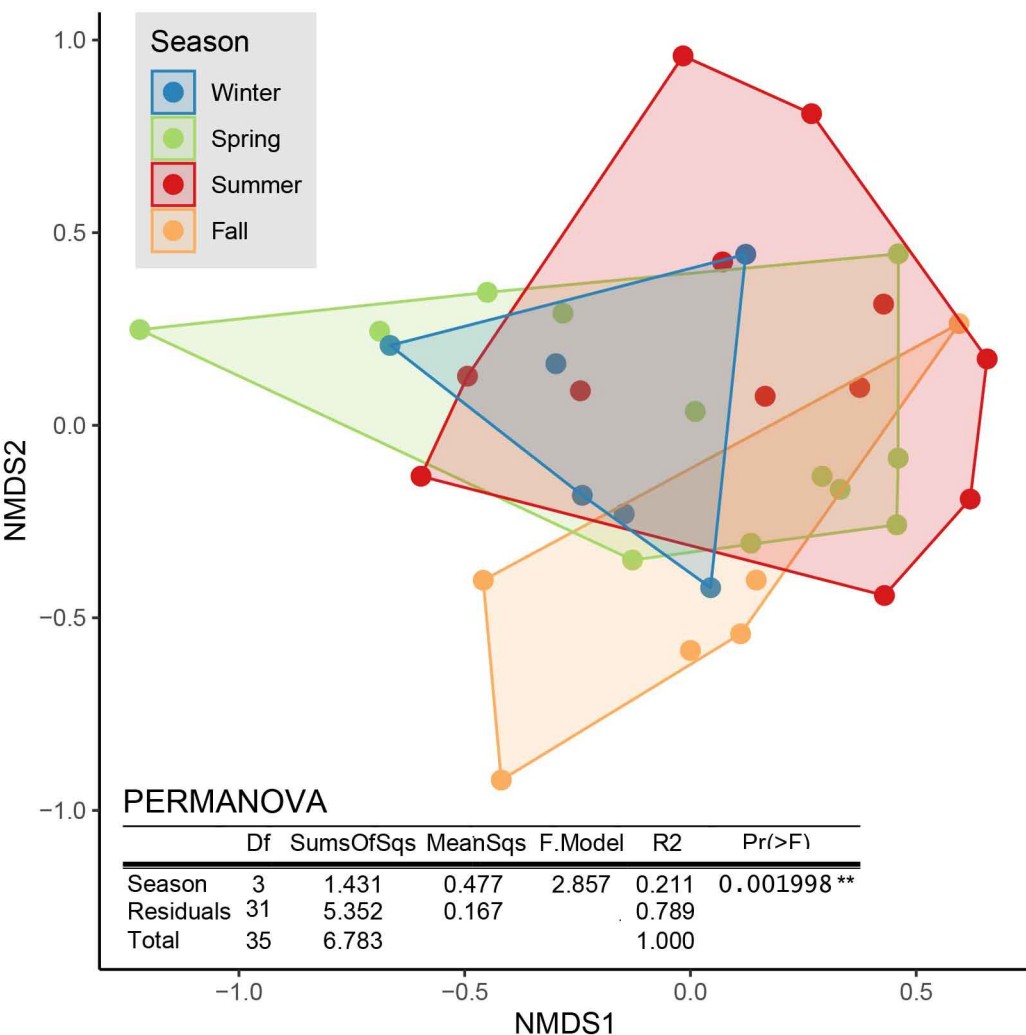

**Fig 6. Ordination graph (NMDS, first two axes) of soil communities based on Bray-Curtis distances (k = 3, stress = 0.174).** The table includes the results of a PERMANOVA testing for differences in taxa composition between seasons based on 1000 permutations.

## Discussion

Values of ACI varied strongly among seasons, with the highest acoustic complexity observed in spring and summer. In these seasons, the diurnal cycle of acoustic complexity was most pronounced. ACI increased in the morning until the early evening and decreased again during the night. The diurnal cycle of soil animals has not been well investigated but these cycles accurately reflect what we know about soil surface dwelling arthropods caught by pitfall traps in different habitats [52, 53]. These dynamics were weaker in fall and disappeared in winter, which seemed to be related to the inactivity period of soil organisms that began in fall and lasted through the winter [54]. Further, higher diurnal variability in ACI in spring and summer might reflect higher variability in microclimatic conditions in these seasons (see next paragraph), which was supported by the strong microclimatic influence on ACI that we found in summer.

Although the responses of soil fauna to microclimatic conditions have been investigated intensively [55, 56], this study is, to our knowledge, the first attempt to investigate the temporal

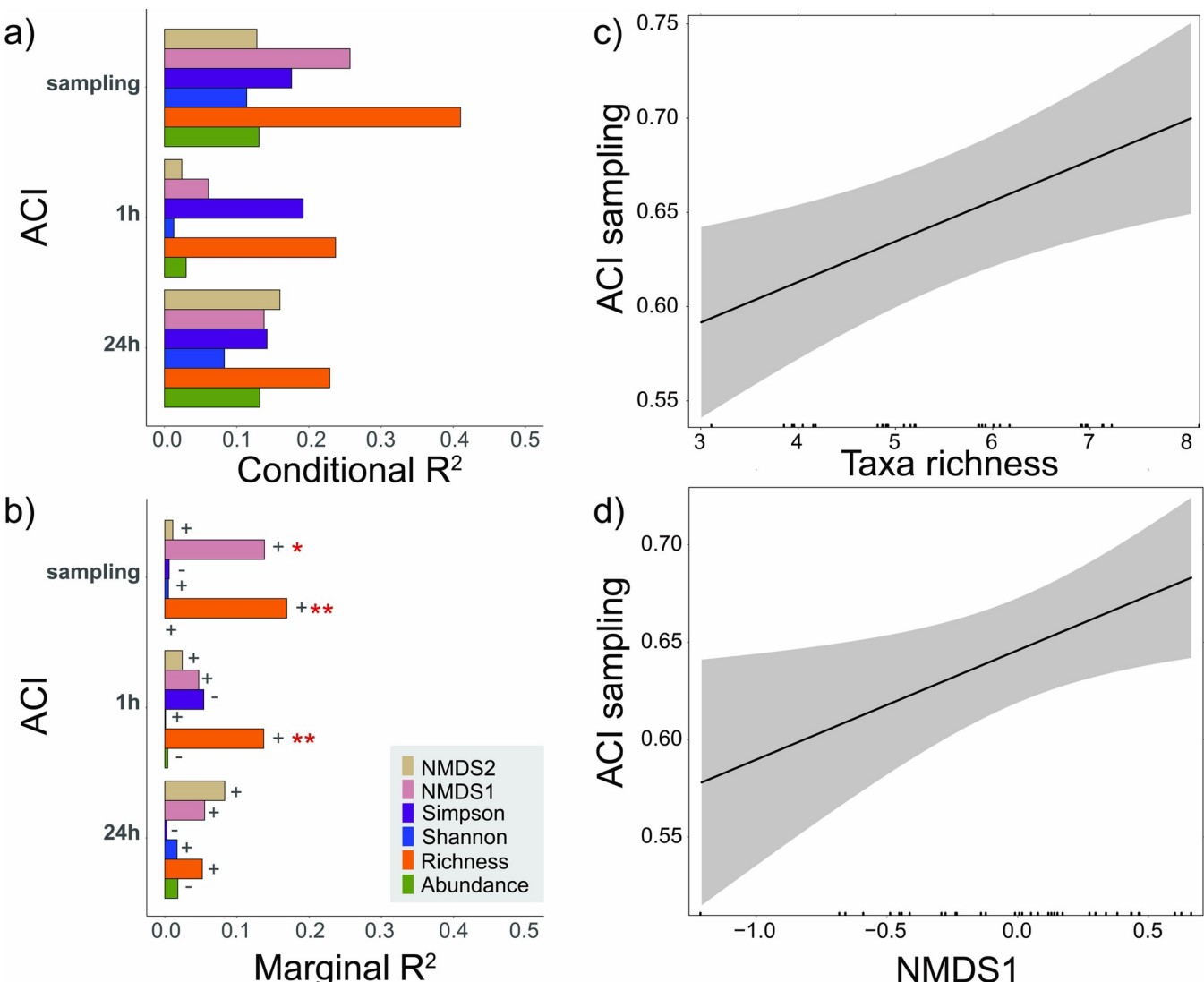

**Fig 7. Relationships between soil community attributes (abundance, taxa richness, Shannon, Simpson, community composition–NMDS 1/2), based on soil cores sampled at sites of different microclimatic conditions on six days in four seasons in 2018 and 2019, and ACI right before sampling, the mean of 1 h around sampling, and 24 h around sampling.** a, b: Conditional and marginal $R^2$-values of linear mixed-effects models (fixed-effect soil community attribute), Significance is indicated by stars (* = p <0.05, ** = p <0.01) and the direction by signs (+, -); c, d: predicted relation between taxa richness / Community composition (NMDS1) and ACI before sampling incl. 95% confidence bands.

dynamics of soil biophonies and geophonies and to relate the acoustic complexity (ACI) of the soil soundscape to the abundance and diversity of important functional groups of soil fauna, the mesofaunas and macrofauna as the primary decomposers and their predators. Our study revealed significant diurnal and seasonal patterns in ACI, which could be partly related to microclimatic conditions. The significant relationship between ACI and soil communities suggests that the spatio-temporal dynamics of soil fauna can be captured by monitoring soil soundscapes.

The dynamics of acoustic complexity were closely related to those of microclimates, where daily increasing surface temperature had the strongest effect on ACI. The ACI peaks at the beginning of soil surface warming, however, seemed to have a strong physical component (i.e., largely belonging to the category of geophonies), as shown by our laboratory experiment with

the artificial heating of soil samples using heat lamps (S7–S9 Figs). These signals could be generated by water evaporating from the surface/litter material [57]. The positive surface temperature effect, by contrast, points to animal activity and diversity: rising soil temperature (at 10 cm depth) provides an increase in the activity and diversity of mesofauna and macrofauna in the uppermost soil layers. This is in line with studies of soil fauna that reported a higher abundance/activity under more open and thus warmer environmental conditions in forests [58]. However, there are limits to this increase, as indicated by the weak negative relationship between soil temperature and ACI. This suggests that when soil temperature reached a critical value, animals fled to deeper, cooler, and moister soil layers [59, 60], and ACI decreased, as acoustic signals were no longer detectable.

We found a significant correlation between ACI measurements and soil animal diversity. Correlations between acoustic complexity and conventional species assessments have been observed only in aboveground ecosystems [61–63]. Thus, we showed for the first time that this might also be transferable to belowground ecosystems. The closer the timing of our audio recordings was to the moment a soil sample was collected, the higher the correlation between ACI and taxa diversity. This suggests that the temporal (and also spatial) variability of activity and composition of the soil animal community was very high, confirming previous studies on the spatio-temporal dynamics of soil fauna [64].

The ACI could be best explained by taxa richness and community composition. This indicates that the occurrence of different taxa is most important for the complexity of soil soundscapes. Different taxa most likely produce different sounds, and thus, the taxa richness rather than the diversity of dominant species best determines acoustic complexity. As among the different q-levels, $q = 0$ (taxa richness) showed the strongest relationship with ACI, rare taxa seem to contribute to the relationship with ACI substantially. Our findings thus suggest that the acoustic complexity also accurately captured the rare taxa in the community, indicating its value for conservation-oriented monitoring programs.

Nevertheless, a few limitations exist in the acoustic observation and measurement of biodiversity in soils (as well as in those above ground). Weather conditions such as rain events make soil recordings difficult or impossible, since raindrops hitting the sensors mask any other signals. This was well illustrated during watering in the irrigation treatment in our study (S12C Fig). Assessments of the biodiversity of soil animals should thus be integrated across longer recording periods of at least several hours or days. Comparisons of acoustic measurements with the actual taxa diversity found in soil samples in our study are subject to certain limitations, because larger and more mobile species, such as ground beetles, could not be found in the samples because they fled before the sample cylinder was driven in. This might also be one reason why the measurements of ACI directly before sampling best correlated with the soil communities. Despite the fact that we measured abundance rather than activity, the high correlation is surprising, as one would expect that not all animals extracted from the soil would be active during acoustic measurements and thus not producing sounds. Therefore, we would expect an even closer relationship if community attributes were measured based on activity rather than density in soil cores. This could be tested using a soil animal sampling device that allows for real-time detection of soil arthropods [65]. Whether species- or even trait-based approaches increase the predictability of soil fauna by measuring soil soundscapes should be addressed in future studies.

In addition to a statistical evaluation of the complexity of animal sounds in soil, it will certainly be important to gain more knowledge about the individual frequency signatures of individual species. The automatic identification and assignment of calling and movement sounds of soil animals, for example, would be an important further step toward a deeper

understanding of the life processes of individual species and the interactions between them, as well as the abiotic influences on soil life [66].

Despite the limitations of our study, the observed high correlation between ACI and taxa richness proves that biodiversity in soil may also be accurately measured and observed acoustically. The passive, acoustic observation of fauna diversity in soil ecosystems offers great potential: On the one hand, it is much easier to implement, both technically and in terms of limited available resources for biodiversity monitoring. The time-consuming collection of soil samples and the expulsion of the animals from the samples, as well as the laboratory analysis, would no longer be necessary. On the other hand, the local ecosystem would not be disturbed by destructive sampling processes.

## Supporting information

**S1 Fig. The acoustic sensors used in the experiment were specially designed for soil recordings.** They consist of a piezo diaphragm from Murata that is 15 mm in diameter and 0.2 mm in thickness. A 10 cm long and 1 mm thick gold-plated copper wire needle was soldered to the back of the brass plate of the diaphragm. The needle functions as a waveguide; it catches the acoustic waves in the ground and passes them onto the piezo element. The diaphragm resonates with the captured sound waves and generates electrical voltage in the electrode on the back side, which is amplified and recorded. The contact microphone is surrounded by a protective plastic housing and is insulated against moisture and short circuits with silicone and an epoxy layer directly on the electrode. A 30 cm coaxial cable leads from the sensor to the preamplifier. To record sounds in the soil, the needle is inserted 10 cm deep into the ground. This allows for the detection of sounds within a radius of approximately 30–100 cm.
(TIF)

**S2 Fig. Soil sampling scheme.** The sampling points were circularly distributed around the acoustic sensor in the middle. Radio of the circle; 50 cm. p = preamplifier, l = light sensor, th = soil temperature and humidity sensor, as = acoustic sensor. Soil samples: s1 = spring 2018, s1.2 = spring 2019, s2 = summer 2018, s2.2 = summer 2019, s3 = fall 2018, s4 = winter 2018.
(TIF)

**S3 Fig. Sampling of soil fauna.** A sample cylinder of 5 cm in diameter and 15 cm length was driven into the soil with a hammer in the vicinity of each acoustic sensor once per season.
(TIF)

**S4 Fig. Berlese apparatus used for extracting soil fauna.** Ordinary light bulbs (40 W) were used and fauna was extracted for 14 days.
(TIF)

**S5 Fig. A qualitative spectral analysis revealed the following general and recurring patterns: The acoustic emissions on the different channels showed a clear distinction between sounds occurring in the soil and aerial sources (control sensor CH 4 in the air).** The soil soundscape showed a characteristic band of acoustic emissions between 100 and 1000 Hz. The emissions within this band consisted of a mix of background noises, such as animals moving in greater distance to the sensor (same emissions as close to the sensor, but with lower amplitude), plant root emissions (resulting from a comparison with emission characteristics of plant roots, described by Gagliano et al. [70], or physical sound sources, such as moving pore water and air (resulting from a comparison with signal characteristics of soil structure alterations and waterfront movements, described by Moebius [71] and Flammer et al. [72]. The most significant and loudest sounds seemed to be movement and feeding noises of animals close to the

sensor (see S5A and S5D Fig channel 3) in frequency bands between 100 and 10'000 Hz. **a)** Spectrogram of the acoustic activity in channels B1–4 on 20.06.2019 12:15 (B1–3/CH 1–3, CH 4 is control in the air). The constant emission band on CH 1–3 shows the spectrum of the soil soundscape. On CH 1 and 3, the spectrogram indicates the movement noises of nearby soil animals. **b)** Sensors B1–4 on 14.06.2019, 15:45. Vibrational calls (CH 2), presumably of a soil insect. Channel 4 in the air shows bird calls. The group of primary decomposers—to a large extent insects and other arthropods that live in and from the litter layer and the uppermost organic soil layers—produce movement and feeding sounds, while the frequencies seem to depend on their body size [73]. Some of these arthropods seem to use the soil matrix as a communication medium. Thus, they produce vibratory sounds with their body or their stridulation apparatus, which propagate over short distances and presumably serve as near-field communication. In the following, a few characteristic examples are highlighted. **c)** Sensors B1–4 on 18.06.2019, 07:05. Stridulation calls (CH 3), presumably by Myrmica rubra or Myrmica ruginodis (observed at the sensor location).
(TIF)

**S6 Fig. Tests with different acoustic indices.** The results of our tests show that no index other than ACI varied over time, and it was even hard to distinguish the daily irrigation periods in the graphs of AEI and AR. Moreover, ACI performed best in resolving daily and seasonal patterns best. To illustrate the temporal dynamics of acoustic complexity at the single measurement spots, heatmaps in a spiral shape were produced. This also allowed detection of specific outstanding events, such as rain or irrigation, in the treatment plot (see S2C Fig). ACI values were mapped with a color scale from dark magenta to light yellow on the spiral graphs below. The higher the ACI, the brighter the color on the spiral graph is represented. Rain and irrigation cause frequent clippings when they hit the acoustic sensors. ACI during rain periods was therefore represented by the brightest colors on the graphs. Similarly, the daily irrigation at the irrigation plot around 23:00 is clearly rendered in yellow. Acoustic diversity in June was recorded by Sensor B1 in the irrigation plot, represented through four acoustic indices: the acoustic evenness index (top left), the acoustic richness (top right), the median of amplitude envelope (bottom left), and the acoustic complexity index (bottom right). Spiral heatmaps of the period 1–29 June 2018; one rotation in the circle represents 24 h. The bright yellow spike shows the irrigation around 23 h.
(TIF)

**S7 Fig. Lab experiments with soil samples taken in the control plot at Pfynwald.** Upper diagram: Untreated sample. Lower diagram: Sterilized sample (frozen at -16˚ C). Shown here is only one of three samples for each treatment because all samples showed similar patterns. To evaluate whether temperature changes in the soil and on its surface generate acoustic emissions of a biotic or abiotic nature, a laboratory experiment was set up. Three soil samples (puncture samples with a diameter and depth of 10 cm) were collected from the control area in the forest and brought cooled to the laboratory. They were subjected to two 1-h heating cycles under infrared heat lamps in a Faraday cage (see S9 Fig). This was done to imitate the falling of sunlight on the forest floor. The three samples were equipped with the same sensors as in the forest, and three days in a row were recorded and measured (see S7 Fig). The three samples were first exposed to heat untreated directly from the field (see S7 Fig, upper diagram). Then, they were sterilized in the freezer at -16˚ C. The sterilization process was probably not complete, but we did not want to destroy the sample's matrix texture by completely freezing it. Individual soil organisms may have survived the freezing process (as the weaker but not completely low ACI curves show in the lower diagram in S7 Fig). The samples were then left to

defrost for 48 h and exposed again to the infrared lamps.
(TIF)

**S8 Fig. Spectrograms of recordings during the heating period in the lab experiment.** Left: untreated soil sample; right: sterilized soil sample. Short high frequent spikes seem to stem from structural changes or evaporating pore water, while broadband signals between 100–1000 Hz seem to originate from soil life.
(TIF)

**S9 Fig. Lab experiments at the Acoustic Ecology Lab, ZHdK.**
(TIF)

**S10 Fig. Cross-correlation function (CCF) between ACI and different microclimate variables (indicated by different colors) for different lags, shown for the five sensors included in the analyses of the relation between microclimate and ACI.** ACI and microclimate variables were differentiated and pre-whitened previously. High correlation at negative lags indicates a strong correlation between ACI and future microclimate, whereas high correlation at positive lags indicates a strong correlation between ACI and past microclimate.
(TIF)

**S11 Fig.** Autocorrelation function (ACF) from residuals of linear mixed-effect models analyzing the effect of (A) daytime and season and (B) microclimate on ACI. Based on these ACFs, the maximum lag was chosen for the autoregressive model in the hierarchical models. Chosen values are indicated by the dashed lines. Note that the time steps of the two models are different (6 h in model A, 10 min in model B).
(TIF)

**S12 Fig. Relationships between microclimatic conditions and acoustic complexity (ACI) over seasons and treatments. a)** Period in spring, 9–12 April 2019; **b)** period in early summer, 23–26 June 2019; **c)** period in mid-summer with active irrigation, 10–13 July 2019.
(TIF)

**S1 Table. Soil samples collected and selected audio recordings for comparison between ACI and the number of taxa.** Comparison tests showed that it is crucial to select recordings made directly before the soil sampling. This is due to the high dynamics in the activity and composition of local soil fauna.
(TIF)

**S2 Table. Sampled taxa and total number of individuals per taxon.** Taxa were identified using [67–69].
(TIF)

**S3 Table. Prior distributions for hierarchical model parameters.**
(TIF)

**S1 File. Sound example. d)** Sensors B1–4 on 21.06.2019, 06:05. Rhythmic feeding/chewing noises (CH 3).
(MP3)

**S2 File. Sound example. e)** Sensors B1–4 on 19.06.2019, 18:25. Rain period. Raindrops hit the acoustic sensors and masked almost every other sounds. The same problem occurs with artificial irrigation. The most typical geophonies in the soil consist of rain hitting the soil and sensors, as well as wind moving plants above ground. These sounds are transmitted into the soil by plant roots. Rain (as well as irrigation) creates such loud noise emissions when it hits the

acoustic sensors that it masks all other sounds and overdrives the recording. In shorter measurement periods or rainy areas, measurements of acoustic complexity must be suspended during rain events.
(MP3)

**S3 File. Sound example. f)** Sensors B1–4 on 14.06., 17:45. Wind/storm event—identifiable on all channels. All sensors respond to strong wind events, since their housing is above ground and the wind moves plants on the surface.
(MP3)

**S4 File. Sound example. g)** Sensors B1–4 on 20.06.2019, 14:55. A jet passes by (bright yellow band at the bottom of the individual channels). Sensors in the soil seem to respond stronger to the airplane's low-frequency noise than the sensor in the air. This indicates that 1. low frequency sound waves enter the soil and are not reflected at the surface and 2. the soil structure starts to resonate with these low frequencies. Anthropogenic noise sources also shape the soundscape in soils. In particular, low-frequency noise, generated by vehicles and roads, is detectable at a distance of several hundred meters. Our station was fortunately far enough away from the highway that ran through the valley. Another anthropophonic source was aircraft noise from military jets. The low frequency thunder of the jets is not reflected at the soil surface; it penetrates it and is clearly measurable there.
(MP3)

**S5 File. Sound example.**
(MP3)

# Acknowledgments

The experiment was carried out in the irrigation experiment of the Swiss Federal Institute for Forest, Snow and Landscape Research WSL in Pfynwald/VS, we thank the responsible persons of the research site, Prof. Dr. Andreas Rigling and Dr. Markus Schaub, for their support. Flurin Sutter from WSL provided us with the necessary map material and Dr. Roman Zweifel provided data sets for the irrigation experiment. Prof. Dr. Michael Stauffacher (ETH USYS TdLab) and Prof. Dr. Rainer Schulin (ETH Institute for Terrestrial Ecosystems) are thanked for their inputs to the design of our experiment; Sabine Lerch and Marilena Schuhmann from the Biovision Foundation for their financial support and cooperation. We thank Stella Mathis and all other interns at the WSL Entomological Laboratory for their support in the analysis of the soil samples. Martin Rüegg is thanked for his assistance with ACI calculations, and Ken Gubler for programming and transports. This study was conducted as part of the research project "Sounding Soil", a collaboration between the Zurich University of the Arts ZHdK/ Institute for Computer Music and Sound Technology, ETH Zurich/USYS TdLab and the Institute for Terrestrial Ecosystems, the Agroscope, Agroecology and Environment/National Soil Monitoring NABO, the Swiss Federal Institute for Forest, Landscape and Snow Research WSL The Research Institute of Organic Agriculture (FiBL) and the Biovision Foundation.

# Author Contributions

**Conceptualization:** Marcus Maeder.

**Data curation:** Marcus Maeder, Xianda Guo, Martin M. Gossner.

**Formal analysis:** Marcus Maeder, Xianda Guo, Felix Neff, Doris Schneider Mathis, Martin M. Gossner.

**Funding acquisition:** Marcus Maeder.

**Investigation:** Marcus Maeder, Xianda Guo, Martin M. Gossner.

**Methodology:** Marcus Maeder, Martin M. Gossner.

**Project administration:** Marcus Maeder.

**Resources:** Marcus Maeder, Doris Schneider Mathis, Martin M. Gossner.

**Software:** Marcus Maeder, Xianda Guo, Felix Neff.

**Supervision:** Marcus Maeder, Martin M. Gossner.

**Validation:** Felix Neff, Martin M. Gossner.

**Visualization:** Marcus Maeder, Xianda Guo, Felix Neff, Martin M. Gossner.

**Writing – original draft:** Marcus Maeder.

**Writing – review & editing:** Marcus Maeder, Felix Neff, Doris Schneider Mathis, Martin M. Gossner.

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
