## [Decision Letter · Decision Letter 0]

1 Dec 2021

PONE-D-21-32125Temporal and spatial dynamics in soil acoustics and their relation to soil animal diversity.PLOS ONE

Dear Dr. Maeder,

Thank you for submitting your manuscript to PLOS ONE. After careful consideration, we feel that it has merit but does not fully meet PLOS ONE’s publication criteria as it currently stands. Therefore, we invite you to submit a revised version of the manuscript that addresses the points raised during the review process.

We look forward to receiving your revised manuscript.

Kind regards,

Haru Matsumoto

Academic Editor

PLOS ONE

Journal Requirements:

"MM - 20'000 CHF - Biovision Foundation - www.biovision.ch - NO."

"NO"

Additional Editor Comments:

Bioacoustics in the soil is an interesting new field. This paper could be a key paper that many future publications would refer to. Having said that I have the same concern with reviewer 2. It is a little stretch to claim that their research is the first one to relate the biodiversity of the soil to soundscape. Both reviewers did a good job suggesting critical references. Please consider including a faunal data set as reviewer 2 suggested which I believe improves the quality of the paper. Overall it is very good paper. Please resubmit with revisions.

Reviewers' comments:

Reviewer's Responses to Questions

**Comments to the Author**

1. Is the manuscript technically sound, and do the data support the conclusions?

Reviewer #1: Yes

Reviewer #2: Yes

2. Has the statistical analysis been performed appropriately and rigorously? 

Reviewer #1: Yes

Reviewer #2: Yes

3. Have the authors made all data underlying the findings in their manuscript fully available?

Reviewer #1: Yes

Reviewer #2: No

4. Is the manuscript presented in an intelligible fashion and written in standard English?

Reviewer #1: Yes

Reviewer #2: Yes

5. Review Comments to the Author

Reviewer #1: The authors conducted acoustic surveys over two years at a Swiss forest site used for irrigation experiments. Samplings of soil biodiversity and microclimatic conditions were performed alongside. The overall aim of the study was to test the hypothesis if soil meso- and macrofauna diversity and composition can be predicted by the complexity of soil soundscapes in both space and time. The study was part of the research project “Sounding soil” of which I have been keen to see scientific results published. Soil acoustic methods have the potential to be a very helpful tool in urgently needed soil biodiversity monitoring, but the available research is quite scarce so far and mainly focused on pest insects. Thus, this manuscript is very timely and it offers very interesting insights into several methodological and ecological aspects of soil acoustic studies. I have some major comments regarding the manuscript, but I hope to see it published soon as it is a very good addition to the currently available soil acoustic literature.

Major comments

I see a bias in the reference list. For me it looks like the authors came from a perspective of acoustics in aboveground ecology and then transferred this perspective to the soil. I am missing references to some relevant soil acoustic studies which someone would cite coming from a belowground perspective (see minor comments for examples).

The discussion has some repetition in the first part (lines 448 – 480). Lines 459 – 463 would actually be a better start sentence for the entire discussion and then the first three paragraphs could be merged and restructured. You write a lot about the relationship between ACI and microclimatic conditions. It sounds kind of general, like you did it for the entire study period,

but you only modeled it for 35 days. Do you think, you would have gotten another result if you modeled it for example for April 2019?

Minor comments

Line 80: Reference 16 doesn’t really refer to the acoustic detection of soil pests. Referencing one of the papers from Mankin et al. or “Zhang et al. (2003): Acoustic estimation of infestations and population densities of white grubs (Coleoptera:

Scarabaeidae) in turfgrass.” would be more fitting here.

Line 82 – 84: I would see the paper of “Mankin et al. (2007): Acoustic Indicators for Mapping Infestation Probabilities of Soil Invertebrates” as already one of the papers going into the direction of acoustically surveying soil animal biodiversity on spatial and temporal scales, although they didn’t target the entire soil fauna.

Lines 117 – 118: Would you still stand by this after looking at, for example, Mankin et al. (2007) and “Inyang et al. (2019): Subterranean Acoustic Activity Patterns of Vitacea polistiformis (Lepidoptera: Sesiidae) in Relation to Abiotic and Biotic Factors”?

Lines 142 – 149: This paragraph would fit better in the introduction.

Line 162: Which soil depth did they cover?

Line 166: Just to make sure I understand it correctly: You recorded the soundscape every 10 minutes for 10 seconds? Why are there 20 seconds in Fig. 2?

Check if all products have a complete reference, e.g. line 176: PlantCare Miniloggers (PlantCare AG, Russikon, Switzerland). (What sensor length did you get for the Miniloggers? ST?)

Line 209: What does ‘selectively’ mean here? Did you just randomly pick out audio recordings or was there still any kind of selection criteria involved?

Line 243: Why does this reduce the influence of the distance between microphones?

Line 249: Is there a reference for Pieretti’s method? What did your own function contribute to calculating the ACI which was not already included in the function acoustic complexity?

Line 294: Which version of ‘rstan’? It is not in the reference list.

Line 303: Why this specific interval of 35 days for the analysis?

Line 362 ff.: What uncertainty measures are shown in the brackets? Looking at Fig. 3, the 95 % is missing as information in the text.

Lines 492f.: “Different taxa most likely produce different sounds…” Do you think different taxa could not only be producers of sound, but also passively affect sound properties in the same way as soil particles do (sound dispersion and attenuation)? See also line 509.

Lines 494 – 496: Really? Don’t quite get the argument here.

Line 499: Did you also have experience with strong wind effects on audio recordings?

Line 520: Possible reference for sound identification: Görres & Chesmore (2019): Active sound production of scarab beetle larvae opens up new possibilities for species-specific pest monitoring in soils

Line 522: You don’t really discuss the limitations of your study.

Lines 623-625: Remove highlighting.

Between reference 54 and 55, there is one more reference.

Reviewer #2: Ths is an original study in soil biodiversity, showing for the first time that soil acoustic diversity is strongly correlated with diversity and species composition of soil fauna. Even though it would have been better to sample more accurately macroinvertebrates (e.g. on a wider surface), I think that we have serious reasons to think that macroinvertebrates were scarcely represented in these dry Scots pine forest stands, but providing soil fauna data would have allowed the reader to better evaluate the quality of faunal sampling. Thus faunal data are required before final acceptance. Another (minor) concern is with the assessment (lines 82-83) that "To our knowledge, there have been no attempts to analyze soil soundscapes across temporal and spatial scales and to relate them to soil animal biodiversity". This not entirely true. Many assays have been done to investigate soil biodiversity by acoustic methods, even if these methods were less sophisticated. Among others, please refer to:

https://www.nature.com/articles/s41598-018-28582-9

https://www.mdpi.com/2571-8789/3/3/45

6. PLOS authors have the option to publish the peer review history of their article (what does this mean?). If published, this will include your full peer review and any attached files.

Reviewer #1: **Yes: **Carolyn-Monika Görres

Reviewer #2: **Yes: **Jean-François Ponge

---

## [Author Response · Author response to Decision Letter 0]

14 Jan 2022

Response to reviewers

Answers Reviewer #1

I see a bias in the reference list. For me it looks like the authors came from a perspective of acoustics in aboveground ecology and then transferred this perspective to the soil. I am missing references to some relevant soil acoustic studies which someone would cite coming from a belowground perspective (see minor comments for examples).

- Many thanks for the reference to the additional literature. We included most of your reference suggestions (see below).

The discussion has some repetition in the first part (lines 448 – 480). Lines 459 – 463 would actually be a better start sentence for the entire discussion and then the first three paragraphs could be merged and restructured. You write a lot about the relationship between ACI and microclimatic conditions. It sounds kind of general, like you did it for the entire study period, but you only modeled it for 35 days. Do you think, you would have gotten another result if you modeled it for example for April 2019?

- We are grateful for this comment as it shows that we were not clear enough in our line of arguments. In the first paragraph of the discussion we focus on seasonal and diurnal patterns and in the second we discuss the relations of ACI to microclimate. To make this clearer we now refer to the second paragraph when mentioning microclimate in the first paragraph (new manuscript line 433).

- It is true that we have only modelled the relationship between ACI and microclimate for a period of 35 days in 2019. However, we have done this deliberately. We selected the time period of highest activity because we expected the clearest relationships in this period, based on our data from 2018. As we discovered in 2018 that microclimate is likely an important factor influencing the soil soundscapes at our study site, we started our microclimate measurements in 2019. We added this information in the methods. It now reads (new manuscript lines 178-179): “This time period was chosen because we expected highest activity of soil fauna during this period based on data from 2018.“ We are quite confident that we would have found the same but weaker relationships in periods of lower activity.

Line 80: Reference 16 doesn’t really refer to the acoustic detection of soil pests. Referencing one of the papers from Mankin et al. or “Zhang et al. (2003): Acoustic estimation of infestations and population densities of white grubs (Coleoptera: Scarabaeidae) in turfgrass.” would be more fitting here.

- Thank you for this helpful suggestion, we replaced the reference with the reference of Zhang et al. (2003).

Line 82 – 84: I would see the paper of “Mankin et al. (2007): Acoustic Indicators for Mapping Infestation Probabilities of Soil Invertebrates” as already one of the papers going into the direction of acoustically surveying soil animal biodiversity on spatial and temporal scales, although they didn’t target the entire soil fauna.

- We revised the sentence to make the novelty of our approach clearer. The sentence now reads (new manuscript , lines 89-92): “In addition, there have been no attempts to analyze soil soundscapes across temporal and spatial scales and to relate them to soil animal biodiversity, although the potential of such research has already been pointed out [18] and first studies on individual soil animal species support this (see references 34/35).”

Lines 117 – 118: Would you still stand by this after looking at, for example, Mankin et al. (2007) and “Inyang et al. (2019): Subterranean Acoustic Activity Patterns of Vitacea polistiformis (Lepidoptera: Sesiidae) in Relation to Abiotic and Biotic Factors”?

- We are aware that such relationships have been studied for individual species, but to our knowledge not studied for meso- and macrofauna at the community level. To make this clearer, we rephrased the sentence (new manuscript, lines 125-129): “There have been first attempts to relate subterranean acoustic activity patterns of individual species to abiotic and biotic factors [34] [35] . However, to our knowledge, the present study is the first to investigate the spatial and temporal dynamics of sounds produced by soil mesofauna and macrofauna at the community level and to relate them to abiotic/microclimatic parameters.”

Lines 142 – 149: This paragraph would fit better in the introduction.

- We agree and integrated this in the Introduction (new manuscript, lines 70-80).

Line 162: Which soil depth did they cover?

- Approx. 10 cm, now noted in the text (new manuscript, line 165): “…covering a soil depth of approx. 10 cm and a volume of approx. 1000 cm3.”

Line 166: Just to make sure I understand it correctly: You recorded the soundscape every 10 minutes for 10 seconds? Why are there 20 seconds in Fig. 2?

- Thank you for noting this, this was actually an error in the text, we recorded for 20s. The text has been changed accordingly (new manuscript, line 169).

Check if all products have a complete reference, e.g. line 176: PlantCare Miniloggers (PlantCare AG, Russikon, Switzerland). (What sensor length did you get for the Miniloggers? ST?)

- Thank you for pointing this out. We now referenced the products correctly (new manuscript, lines 177-180). 

Line 209: What does ‘selectively’ mean here? Did you just randomly pick out audio recordings or was there still any kind of selection criteria involved?

- We selected the audio files randomly. We now clarify this (new manuscript, lines 208- 210): “This was done by listening to randomly selected audio recordings of different times of days, months and seasons.”

Line 243: Why does this reduce the influence of the distance between microphones?

We think that this is a misunderstanding. The method reduces the influence of the distance between the microphone and the recorded organism (if very close, sum(I) is very large and ACI is reduced accordingly). We have changed the text slightly to make this clearer (new manuscript, lines 244-245).

Line 249: Is there a reference for Pieretti’s method? What did your own function contribute to calculating the ACI which was not already included in the function acoustic complexity?

- We apologize for not citing the method of Pieretti. The reference is is now added to the manuscript (new manuscript, line 249). We closely followed this method and just adapted it to our data.

Line 294: Which version of ‘rstan’? It is not in the reference list.

- The reference including version number was now added to the reference list.

Line 303: Why this specific interval of 35 days for the analysis?

- Microclimatic variables were only assessed during the growing season 2019 because we discovered in 2018 that microclimate might be an important factor influencing soundscapes at the study site. The period of expected highest activity of soil fauna was chosen. We clarify this in the revised text (new manuscript, lines 301-303).

Line 362 ff.: What uncertainty measures are shown in the brackets? Looking at Fig. 3, the 95 % is missing as information in the text.

- Thank you for pointing this out. These are 95% highest density intervals, the information has been added to the text (new manuscript, line 362).

Lines 492f.: “Different taxa most likely produce different sounds…” Do you think different taxa could not only be producers of sound, but also passively affect sound properties in the same way as soil particles do (sound dispersion and attenuation)? See also line 509.

- No, we don’t think that this is the case since the soil faunal biomass makes a very small physical part of the soil matrix, maybe with the exception of a mass occurrence of soil pest species or an ant nest nearby the acoustic sensors for example. However, this was neglectable in our case.

Lines 494 – 496: Really? Don’t quite get the argument here.

- We analyzed the taxa diversity based on different q-levels. With increasing q-levels from 0 (taxa richness) to 1 (taxa Shannon diversity) to 2 (taxa Simpson diversity) dominant taxa are more strongly weighted. As q=0 showed the strongest relationship with ACI the rare taxa seem to contribute to this relationship substantially. We made this clearer in the discussion by rephrasing this part (new manuscript, lines 471-474): “As among the different q-levels, q=0 (taxa richness) showed the strongest relationship with ACI, rare taxa seem to contribute to the relationship with ACI substantially. Our findings thus suggest that the acoustic complexity also accurately captured the rare taxa in the community, indicating its value for conservation-oriented monitoring programs.”

Line 499: Did you also have experience with strong wind effects on audio recordings?

- Yes, wind also affected audio recordings (see Supporting Information S5f Fig), but did not affect our main finding. Strong wind events occurred rarely and were statistically not significant.

Line 520: Possible reference for sound identification: Görres & Chesmore (2019): Active sound production of scarab beetle larvae opens up new possibilities for species-specific pest monitoring in soils.

- Thank you for this useful reference, it has been integrated in the text (revised document, line 550).

Line 522: You don’t really discuss the limitations of your study.

- Here we disagree with the reviewer. The limitations of our study are discussed in line 476-492.

Lines 623-625: Remove highlighting.

- This has been done.

Between reference 54 and 55, there is one more reference.

- Thank you for pointing us on this. It’s a redundant reference which has been removed.

Answers Reviever #2

This is an original study in soil biodiversity, showing for the first time that soil acoustic diversity is strongly correlated with diversity and species composition of soil fauna. Even though it would have been better to sample more accurately macroinvertebrates (e.g. on a wider surface), I think that we have serious reasons to think that macroinvertebrates were scarcely represented in these dry Scots pine forest stands, but providing soil fauna data would have allowed the reader to better evaluate the quality of faunal sampling. Thus faunal data are required before final acceptance. 

- We appreciate the general positive evaluation of our manuscript. Regarding the faunal data requested by the reviewer we are not completely sure what is meant. We provided a table in the Supporting Information (Table S2. Sampled taxa and total number of individuals per taxon) were we show the taxonomic level on which we identified soil fauna and in addition give the total abundance of all taxa. 

Another (minor) concern is with the assessment (lines 82-83) that "To our knowledge, there have been no attempts to analyze soil soundscapes across temporal and spatial scales and to relate them to soil animal biodiversity". This not entirely true. Many assays have been done to investigate soil biodiversity by acoustic methods, even if these methods were less sophisticated. 

- Based on the comments of reviewer 1 we added a few references of studies that addressed the relationship of soil acoustics and single species (see response to comments of reviewer 1). However, we for the first time address the relationship of soil soundscapes and soil biodiversity at the community level. We made this clearer in our new manuscript (lines 87-93).

- Regarding your specific suggestions:

Among others, please refer to:

https://www.nature.com/articles/s41598-018-28582-9

- Also in this paper the authors do not address soil biodiversity at the community level, but focus on earthworms exclusively. We added this paper to our introduction (new manuscript, line 87). 

https://www.mdpi.com/2571-8789/3/3/45

- We already cited this paper (line 84 in the original version, line 91 in the new manuscript). This is however rather a perspectives paper than based on an empirical study.

---

## [Editor Report · Decision Letter 1]

24 Jan 2022

Temporal and spatial dynamics in soil acoustics and their relation to soil animal diversity

PONE-D-21-32125R1

Dear Dr. Maeder,

We’re pleased to inform you that your manuscript has been judged scientifically suitable for publication and will be formally accepted for publication once it meets all outstanding technical requirements.

Kind regards,

Haru Matsumoto

Academic Editor

PLOS ONE
---

## [Editor Report · Acceptance letter]

31 Jan 2022

PONE-D-21-32125R1 

Temporal and spatial dynamics in soil acoustics and their relation to soil animal diversity 

Dear Dr. Maeder:

I'm pleased to inform you that your manuscript has been deemed suitable for publication in PLOS ONE. Congratulations! Your manuscript is now with our production department. 

Kind regards, 

on behalf of

Dr. Haru Matsumoto 

Academic Editor

PLOS ONE